# Utilizing Design Thinking for Effective Multidisciplinary Diabetes Management

**DOI:** 10.3390/healthcare11131934

**Published:** 2023-07-04

**Authors:** Ming-Chen Hsieh, Yu-Ming Kuo, Yu-Lun Kuo

**Affiliations:** 1Department of Medical Education, Hualien Tzu Chi Hospital, Buddhist Tzu Chi Medical Foundation, Hualien 97002, Taiwan; 2School of Medicine, Tzu Chi University, Hualien 97004, Taiwan; 3Department of Marketing and Distribution Management, Tzu Chi University of Science and Technology, Hualien 970302, Taiwan; ss248@ems.tcust.edu.tw; 4Department of Nursing, Tzu Chi University of Science and Technology, Hualien 970302, Taiwan; ss246@ems.tcust.edu.tw

**Keywords:** design thinking, discharge planning, interprofessional education

## Abstract

(1) Background: Design thinking, as a human-centered design method, represents a unique framework to support the planning, testing, and evaluation of new clinical spaces for diabetic care throughout all phases of construction. This approach prioritizes the needs and experiences of diabetic patients to create innovative and effective healthcare environments. By applying design-thinking principles, healthcare facilities can optimize the design and functionality of their clinical spaces, ensuring a patient-centered approach to diabetic care. This holistic and personalized approach can ultimately enhance the overall quality of diabetic care provided to patients. (2) Methods: The study used the action research method and progressively explored diabetes patients’ needs and preferences for care, subsequently developing creative solutions to achieve the goals. There were six doctors, seven nursing staffs, four case managers and three family members who participated in the design-thinking workshop. (3) Results: The participating trainees in this study developed unique and innovative solutions during the iterative process of “divergent thinking” and “focused thinking”, including diabetes dietary guidelines for food ordering and delivery platforms, and the design of accompanying health-education picture books to enable patients to learn the care process and precautions before, during, and after discharge. (4) Conclusions: This continuing education model promoted sharing among participants, improved collaboration and mutual learning, and increased motivation through goal setting.

## 1. Introduction

The healthcare landscape has witnessed a growing emphasis on collaboration and patient-centered care. It is essential to recognize the unique challenges faced by individuals with diabetes and prioritize their specific needs in healthcare delivery. By fostering collaboration among healthcare professionals, including patients and their families, innovative approaches can be developed to enhance diabetes care and improve patient outcomes [1]. Discharge planning (DP) services play a crucial role in enhancing the quality of care [2]. DP services are essential in ensuring a smooth transition for patients from the hospital to their post-acute care settings. DP involves comprehensive case management and interprofessional collaboration, necessitating effective communication and coordination among healthcare professionals to tailor discharge plans to the specific needs and circumstances of individual patients. One of the key aspects of DP is the identification and assessment of patients’ medical, physical, and psychosocial needs prior to discharge. This involves thorough evaluations by healthcare professionals, including physicians, nurses, social workers, and therapists, to determine the appropriate level of care and support required by each patient [3]. Professionals from different backgrounds may offer advice and guidance based on their own professional perspectives. However, it is common for these perspectives to lack integration and may even lead to conflicting situations. Therefore, interprofessional education (IPE) plays a pivotal role in fostering effective collaboration, interdisciplinary communication, and synergy among healthcare professionals, consequently fortifying and bolstering the healthcare system. By engaging in IPE, healthcare professionals have the opportunity to cultivate a shared understanding of each other’s roles, responsibilities, and expertise, facilitating the development of mutual respect and trust within the team [4]. Traditionally, hospitals have utilized “case discussions” as a means to facilitate IPE [5]. This approach focuses on individual expertise rather than understanding the roles of others and fostering cooperative learning and interaction [6]. This study suggests that the educational approach of design thinking is more effective in enhancing the effectiveness of collaborative teamwork than case discussions, allowing professionals, patients, and their families from various disciplines to brainstorm and promote patient-centered care methods. In the following paragraphs of this study, we will first introduce the connotation of design thinking, explaining that it is a process that utilizes the unique perspectives, experiences and skills of each participant through cross-disciplinary cooperation and continuous improvement. This study uses the action research method and actually invites doctors, nursing staff, case managers and family members to conduct design-thinking workshops. The aim of the results of this research is to find out an innovative and patient-centered discharge care model [7].

### 1.1. Design-Thinking Content

Design thinking is widely recognized as a strategic approach that enables organizations to gain a competitive advantage by formulating solutions that effectively align with the evolving needs and preferences of their service users. Emphasizing a human-centered and iterative process, design thinking fosters empathy, creativity, and collaboration in a multidisciplinary way, facilitating the identification of unmet service users’ needs, the exploration of innovative ideas, and the development of viable solutions. By integrating service users’ insights and employing a systematic problem-solving framework, organizations can enhance their ability to deliver products, services, and experiences that resonate with their target audience, ultimately positioning themselves ahead of their competitors in the dynamic marketplace [8]. The approach encompasses the application of empathetic understanding, the consideration of diverse perspectives, the generation of multiple conceptual alternatives, the utilization of iterative problem-solving methodologies, the validation of potential solutions through prototyping, and the assimilation of insights gained from errors and failures, all aimed at fostering the creation of a solution that is centered on human needs and values. By embracing this comprehensive approach, organizations can effectively navigate complex challenges, optimize the quality and relevance of their solutions, and ultimately enhance overall user experience and satisfaction.

An essential difference between design thinking and other problem-solving methods is that design thinking focuses on creating solutions, whereas most other methods emphasize solving the causes of problems [9]. It is crucial to go beyond the framework of finding the right solution using analytical thinking and to develop effective solutions by introducing new options. Design thinking shifts risk avoidance and structured process adoption to an iterative work style that encourages risk-taking and learning from mistakes and that tests out many solutions [10]. Huang et al. [11] examined the impact of a design-thinking approach in an interprofessional education program focused on a human sexuality course. Cleckley et al. [12] detailed the use of a design-thinking process to identify discrepancies between expected interprofessional education competencies, curriculum implementation, and interprofessional collaboration. Gomutbutra et al. [13] discovered that both medical students and technology students perceived design thinking as beneficial to their careers and expressed satisfaction with its application for enhancing interprofessional education between technology and medicine for innovative elder pain care.

### 1.2. Design-Thinking Process

The design-thinking process consists of five fluid steps: empathize, define, ideate, prototype, and test. The fundamental principle is to create solutions based on the needs of both “users” and “stakeholders” [14]. In the realm of medical services, users may refer to patients or medical personnel who encounter difficulties, whereas stakeholders could be the family members of users, physicians, and individuals from other professions who are familiar with the challenges.
Empathize

The initial stage of the design-thinking process, known as the empathize step, serves two primary objectives: firstly, to help innovators gain a comprehensive understanding of users’ experiences and persistent problems with the service that can inconvenience them; and secondly, to ascertain the level of interest among users and stakeholders in supporting the development of the proposed product or idea [15]. The empathize step involves exploring the ideas, values, emotions, and practices related to the identified problem and creating a framework for a solution. This is achieved through interviews and observations of user behavior. It is important to use techniques that promote open-ended and non-judgmental interactions to gain a thorough understanding of users’ needs. This step helps uncover overlooked issues and ensures that the problem is accurately defined for the subsequent problem-solving stage.
2.Define

Following the identification of users’ or stakeholders’ persistent problems, the subsequent stage is the define phase, which serves as a definitive reference point for the medical team to revisit and reassess throughout various stages of implementation [16]. Health professionals can utilize specific case scenarios that align with the feedback received from users or stakeholders to illustrate the issue at hand during this stage [17]. After defining the problem, users or stakeholders can be consulted to validate the accuracy of the identified problem statement. In the Define Stage, there is no need to ask experts to directly provide an action plan; rather, the aim is to enter the Ideate Stage and find more action plans through collective creative thinking.
3.Ideate

The ideate stage plays a pivotal role in the design-thinking process, as it involves the generation of a wide range of potential solutions to address the defined problem [15]. Iterative and incremental idea development is essential, building upon the established problem statement. Cross-disciplinary team members should contribute their insights to foster diverse perspectives and increase the likelihood of identifying an optimal solution. It is crucial to keep an open mind and consider all ideas. Revisiting previous stages if necessary ensures a comprehensive understanding of the problem. Before moving to the next phase, the team should narrow down the potential solutions to a few options, avoiding excessive divergence.
4.Prototype

The prototyping step involves creating a tangible representation of the product or solution for further development. It should be cost-effective and efficient, focusing on collecting feedback from users through visual presentations. In design-thinking education, inexpensive materials, such as popsicle sticks, pipe cleaners, tape, and colored pens, are often used for rapid prototyping, fostering resourcefulness and creative problem-solving. This approach promotes inclusive and practical learning, encouraging innovation and hands-on exploration within design-thinking education [18]. The minimum requirement is to allow for tactile or visual feedback from users to generate valuable comments for improving the solution. Interactions, procedures, or demonstrations can also be used to simulate problems and test potential solutions.
5.Test

During the testing phase, users and stakeholders actively participate in providing feedback to assess the extent to which the initial persistent problems and needs have been effectively addressed [15]. To mitigate risks in resource allocation, it is important to identify potential factors that may hinder the achievement of desired goals. Users should be given ample time to evaluate the proposed solution and compare prototypes, ask questions, and explore alternative options. The project team should document user feedback and make necessary adjustments, continuously reassessing the outcomes of the empathize and define stages to refine and optimize the solution.

## 2. Materials and Methods

Our study assesses the effectiveness of the 2020 diabetes care network staff continuing education program. This one-day, eight-hour program was held at a teaching hospital in eastern Taiwan. We fully implemented the five steps of design thinking. The final design-thinking result is to invite the Deputy Director of the Nursing Department and family members to evaluate whether the plan is feasible and valuable. We hope that the proposals proposed by various groups can be used to improve the healthcare of diabetes patients in the future.

We used action research as the research method. Action research is a research method that combines investigation and problem-solving simultaneously [19]. In the field of education, it refers to systematic research that aims to bring about change and improvement in educational practice [20,21,22]. In the context of medical education, this research adopts the action-research method to address a specific problem and enhance teaching. The focus of action research is not on developing theory or generalizable research findings but on de-signing research that addresses the characteristics of a specific problem in a particular time, place, and situation, while incorporating relevant educational theories [20]. Action research follows a systematic and cyclical inquiry process of planning, action, observation, and reflection. The following are the scenarios and steps for using action research in this study.

The continuing education program was led by three instructors and involved the participation of twenty individuals. We invited six doctors, seven nursing staffs, four case managers and three family members to join this workshop. All participants were divided into five teams. Throughout the activity, participants engaged in numerous discussions, documenting their ideas on sticky notes or posters. Table 1 shows the key activities and tools utilized for each of the design stages. We have preserved all these files as part of our study. Furthermore, we diligently recorded the shared content from each team during the event. To gather additional insights, we developed a reflection questionnaire to capture participants’ thoughts and suggestions regarding their participation. These materials serve as valuable research text, enabling us to analyze and draw conclusions from the study.

## 3. Results and Discussion

In the Empathize stage, we utilized the Persona tool, as outlined in previous studies [16]. During this phase, we facilitated group discussions with the participants, encouraging them to collectively create visual representations of diabetic patients on poster paper. This involved drawing representative features and documenting key physical and behavioral traits. By employing this approach, the participants were able to center their attention on the specific individuals requiring assistance, fostering a shared understanding of their needs. Furthermore, this activity facilitated the establishment of clear goals and the formation of common norms within each group.

In the Define stage, we employed the Rose, Bud, and Thorn method, as documented in prior literature [17]. Participants were tasked with identifying the positive and negative factors that exerted an influence on the self-care practices of diabetic patients within their home environments, with a particular emphasis on identifying modifiable factors. Each participant was provided with post-it notes in three distinct colors: red for positive factors, blue for negative factors, and green for modifiable factors. Within a time frame of 10 min, participants recorded relevant factors on the post-it notes. Subsequently, the groups were allotted 20 min to share their lists and collaboratively categorize the items into coherent themes. These themes were then visualized through the creation of mind maps. The objective of this activity was to discern the contextual factors that significantly impact diabetic patients’ daily self-care routines at home. By elucidating these factors, which can either fortify or undermine patients’ health behaviors, a foundation for subsequent creative ideation was established.

The Ideate stage encompassed two distinct steps. In the first step, the groups were directed to select a contextual factor for improvement, and each participant formulated five sentences following the structure of “How might we do X to achieve Y?” Through this process, each group generated a total of 20 problem statements that required a resolution. Subsequently, the groups were tasked with identifying the most crucial problem statement to be transformed into actionable items. The objective of this activity was to break down significant ideas into smaller, manageable projects, optimizing the utilization of limited resources, labor, and time [18].

During the second step, every participant was provided with a sheet of paper comprising 15 vacant spaces and instructed to generate five distinct solutions within these spaces. Following this, the paper was passed to another member within the group, who then added their own set of five unique solutions. This iterative process aimed to stimulate creative thinking among the participants and encourage diverse perspectives. As a result of this collaborative ideation exercise, each group collectively amassed a total of 60 solutions.

During the Prototype stage, each group engaged in the process of selecting the most viable and valuable solution from the pool of 60 generated solutions, with the aim of creating a well-defined project design. To determine the feasibility and value of each solution, we guided the groups to categorize them using an “importance × difficulty” model. Participants were instructed to construct a 2 × 2 table on a poster paper, with the x-axis representing importance and the y-axis representing difficulty. We encouraged the groups to prioritize solutions that exhibited a high importance and low difficulty, as these would yield immediate results and attract organizational investment in terms of resources, labor, and time. Solutions falling into the high-importance, high-difficulty quadrant were acknowledged for their strategic value but were anticipated to require greater efforts in terms of garnering institutional support. Given that several solutions fell within the same quadrant, we prompted the groups to further categorize and refine their solutions, thereby fostering creative thinking. Once a solution was selected, the groups proceeded to create a prototype in the form of a storyboard. For instance, one group proposed a service model for self-health management, which was effectively communicated through the presentation of a comic strip that elucidated the mechanics of the model and illustrated the anticipated outcomes.

In the Test stage, each group conducted presentations of their respective posters for fellow participants, opting not to use computerized slides. Physicians and diabetic patients were invited to attend these presentations and provide valuable feedback, aimed at assessing the feasibility and value of the proposed designs. This stage sought to gather expert and user opinions as a reference for subsequent design enhancements or as a foundation for integrating diverse designs into a comprehensive solution. Throughout this process, participants acquired insights into interdisciplinary services pertaining to diabetes discharge preparations, encompassing areas such as insulin injection health education, nutrition education, physical activity, public health, and the self-monitoring of blood glucose.

The final outputs of the ideation projects encompassed a range of innovative solutions.

The overwhelming response from participants indicated that the content generated through the continuing education ideation projects resonated closely with the needs of diabetic patients, attesting to the relevance and efficacy of the developed solutions. These solutions consisted of personalized dietary recommendations specifically designed for diabetes management on food ordering and delivery platforms. Additionally, a visually captivating health-education picture book incorporating dynamic character dialogues and audio animations was developed. Furthermore, a proposal was presented for the production of a health-education short film, complemented by both physical and electronic books. These mediums were strategically designed to deliver informative messages to patients and their families in captivating ways, offering comprehensive guidance throughout the patient-care process before, during, and after discharge. The results indicated the potential of these solutions to enhance patient engagement and improve the effectiveness of diabetes care. The overwhelming response from participants indicated that the content generated through the continuing education ideation projects resonated closely with the needs of diabetic patients, attesting to the relevance and efficacy of the developed solutions.

## 4. Conclusions

Through our research, we have observed that the instructional approach of design thinking offers valuable opportunities for individuals from diverse professional backgrounds to engage in interaction and communication. By employing design-thinking methodologies and engaging in meaningful discussions, participants can gain insights into the specialized knowledge of various disciplines, thereby developing an understanding of the significance of each specialty within the context of patient care. These collaborative efforts have yielded fruitful outcomes and unleashed the creative potential of employing design-thinking strategies in problem-solving endeavors. Design thinking is a methodology that generates innovative solutions for complex problems, whereas case studies are research methods that are used to examine specific instances or phenomena in detail. Both approaches offer valuable insights and have their unique strengths in different research and problem-solving contexts.

Our research can be used to illustrate that the process of problem-solving should not be pursued in isolation; instead, it necessitates the cultivation of interdisciplinary literacy and efficient teamwork. Engaging in cross-domain dialogues can foster the generation of diverse solutions and elevate the quality of diabetes care to new heights. Only through mutual engagement, comprehension of the expertise and values of other fields, the dismantling of traditional hierarchical barriers, and the establishment of a robust team mindset can an optimal division of labor and seamless collaboration be achieved, ultimately leading to the provision of comprehensive and error-free medical care for patients.

Our research aligns with previous studies in the literature, which have also shown that employing design-thinking methodologies is effective in fostering interdisciplinary collaboration, promoting creative problem-solving, and improving patient-care outcomes. For example, in healthcare and education, design thinking has been found to enhance interprofessional education programs and improve educational practices [11,12,13]. These studies emphasize the importance of engaging diverse stakeholders, encouraging cross-disciplinary dialogue, and integrating design-thinking principles to address specific challenges and develop innovative solutions.

We finally summarize two possible methods: personalized dietary recommendations for diabetes management on food ordering and delivery food platforms. While these methods require further implementation and validation to be effective, the process itself has been invaluable. It has allowed participants to identify user needs, explore problems from diverse perspectives, and collaboratively reach consensus during problem-solving. As a result, we strongly believe that design thinking is highly suitable for medical education, as it fosters empathy and enhances problem-solving abilities. By employing design-thinking principles, we can cultivate a more patient-centered approach to healthcare.

## Figures and Tables

**Table 1 healthcare-11-01934-t001:** The key activities and tools utilized for each of the design stages.

Design Stage	Key Activities	Tools Used
Empathize	- Group discussions creating visual representations	Poster paper, drawing materials
	- Establishing clear goals and common norms	
Define	- Rose, Bud, and Thorn method for identifying factors	Post-it notes (red, blue, green)
	- Categorizing factors into themes and creating mind maps	
Ideate	- Formulating problem statements	Sentence structure guidelines
	- Generating multiple solutions	Individual sheets of paper
Prototype	- Selecting viable solutions	Importance × difficulty model
	- Creating well-defined project design	Poster paper, storyboard
Test	- Presentations and gathering feedback	Posters, presentations, expert opinions

## Data Availability

The data used to support the findings in this study are available from the corresponding author upon request.

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
