# Peer review of "Utilizing Design Thinking for Effective Multidisciplinary Diabetes Management"

_healthcare, 2023, doi:10.3390/healthcare11131934_

Round 1

Reviewer 1 Report

In this manuscript, Hsieh et al. used the "design thinking" process for effective diabetes management, including "empathize", "define", "ideate", "prototype", and "test".  This report can have important implications to generate innovative ideas and promote cross-disciplinary communication. The paper is comprehensive. I only have a few minor comments:

- line 16-18, "This study uses..." this sentence seems should be moved to "Methods" rather than "Background".

- line 43, "1.1 Design-thinking content and process" This subtitle is the same as line 56 "1.2 Design-thinking content and process".

The English in this manuscript is good to me. I have no comments on the quality of English language.

Author Response

Response to Reviewer 1 Comments

Point 1:

- line 16-18, "This study uses..." this sentence seems should be moved to "Methods" rather than "Background".

Response 1: Thank you for your feedback. We have revised the abstract accordingly. Please find the modified version line 15-20.

Point 2: - line 43, "1.1 Design-thinking content and process" This subtitle is the same as line 56 "1.2 Design-thinking content and process".

Response 2: Thank you for your comment. We have revised the subtitle as per your suggestion. Please see the modified version at line 61 and 87

Reviewer 2 Report

The abstract needs to be more informative about the methods and chosen population.

The introduction is confusing; first, it talks about interprofessional collaboration, then talks about including patients and family members in decision-making as professionals, and finally talks about applying design thinking. The aim of the study is not clear based on what is said in the introduction.

The method needs much more detail. It is not clear who were the participants and what was the education material, or how long they were educated,..?

The results section continues to talk about methods. Also, there is no statistical analysis to support the findings.

The conclusion is not based on what was presented in the results.  

There are some minor grammatical errors. The paper should be proofread.

Author Response

Response to Reviewer 2 Comments

Point 1: The abstract needs to be more informative about the methods and chosen population.

Response 1: We have re-written the abstract, specifically addressing the points raised in your comment. From line 21 – 24 : (2) Methods: The study used the action research method and progressively explored the diabetes patients' needs, preferences for care and subsequently developed creative solutions to achieve the goals. There are six doctors, seven nursing staffs, four case managers and three family members to participate design thinking workshop.

Point 2: The introduction is confusing; first, it talks about interprofessional collaboration, then talks about including patients and family members in decision-making as professionals, and finally talks about applying design thinking. The aim of the study is not clear based on what is said in the introduction.

Response 2: We have re-written the introduction.

From line 47-68: Professionals from different backgrounds may offer advice and guidance based on their own professional perspectives. However, it is common for these perspectives to lack integration and may even lead to conflicting situations. Therefore, Interprofessional education (IPE) plays a pivotal role in fostering effective collaboration, interdisciplinary communication, and synergy among healthcare professionals, consequently fortifying and bolstering the healthcare system. By engaging in IPE, healthcare professionals have the opportunity to cultivate a shared understanding of each other's roles, responsibilities, and expertise, facilitating the development of mutual respect and trust within the team [4]. Traditionally, hospitals have utilized "case discussions" as a means to facilitate IPE. [5]. This approach focuses on individual expertise rather than under-standing the roles of others and fostering cooperative learning and interaction [6]. This study suggests that the educational approach of design thinking is more effective in enhancing the effectiveness of collaborative nursing than case discussions, allowing professionals, patients, and their families from various disciplines to brainstorm and promote patient-centered care methods. In the following paragraphs of this study, we will first introduce the connotation of design thinking, explaining that it is a process that utilizes the unique perspectives, experiences and skills of each participant, through cross-disciplinary cooperation and continuous improvement. This study uses the action research method and actually invites doctors, nursing staff, case managers and family members to conduct design thinking workshops. The research result is to find out an innovative and patient-centered discharge care model. [7].

Point 3: The method needs much more detail. It is not clear who were the participants and what was the education material, or how long they were educated,..?

Response 3: We appreciate your insightful reminder. We have restructured the method section, providing more detailed information about the process. We reinforce the paragraphs and explanations of our research methods.

From line 163-188: We used action research as research method. Action research is a research method that combines investigation and problem-solving simultaneously [19]. In the field of education, it refers to systematic research that aims to bring about change and improvement in educational practice [20]. In the context of medical education, this research adopts the action research method to address a specific problem and enhance teaching. The focus of action research is not on developing theory or generalizable research findings but on designing research that addresses the characteristics of a specific problem in a particular time, place, and situation, while incorporating relevant educational theories [20]. Action research follows a systematic and cyclical inquiry process of planning, action, observation, and reflection. The following are the scenarios and steps for using action research in this study.

The educational course was led by three instructors and involved the participation of twenty individuals. We invites six doctors, seven nursing staffs, four case managers and three family members to conduct design thinking workshops. In addition to comprehensive lectures, the training incorporated a project that entailed the design of diabetes discharge preparation services. For this project, participants were divided into five teams. We conduct workshops using the five major processes of Design Thinking from Stanford University. Towards the conclusion of the course, feedback on the projects was solicited from both patients and medical specialists.

The diabetes network was chosen for this study due to the significance of diabetes care and management in healthcare. Comprised of healthcare professionals and family members, the network aimed to address challenges in diabetes care through the use of design thinking. Design thinking offered a systematic and collaborative approach to promote innovation, problem-solving, and patient-centered solutions. By applying design thinking principles, the network aimed to foster interdisciplinary collaboration, identify unmet patient needs, develop creative solutions, and enhance the quality of diabetes care and education. Please refer to the revised text at the indicated line 182-189.

Point 4: The results section continues to talk about methods. Also, there is no statistical analysis to support the findings.

Response 4: Although action research tends to be qualitative in nature, we have incorporated the use of table to provide a clearer understanding of our research steps and methods. These tables serve as visual aids to present the key activities and tools employed in each design stage, enhancing the transparency and comprehensibility of our research process.From line 271-273

Point 5: The conclusion is not based on what was presented in the results. 

Response 5: We have added the conclusion.

From line 274-290: Our research aligns with previous studies in the literature, which have also shown that employing design thinking methodologies is effective in fostering interdisciplinary collaboration, promoting creative problem-solving, and improving patient care outcomes. For example, in healthcare and education, design thinking has been found to enhance interprofessional education programs and improve educational practices [11, 12 ,13]. These studies emphasize the importance of engaging diverse stakeholders, encouraging cross-disciplinary dialogue, and integrating design thinking principles to address specific challenges and develop innovative solutions.

Reviewer 3 Report

1. No sufficient Literature review.

2. State-of the art comparison is not present in study.

3. Paper is just like theory, no experimentation is visible at first sight.

4. Presentation is not good.

Overall paper is not acceptable in this form. Updations are required in it. 

Language & Grammar improvement is needed

Author Response

Response to Reviewer 3 Comments

Point 1: No sufficient Literature review.

Response 1 : We add the literature review.From line 94-101: Huang et al. [11] examined the impact of a design thinking approach in an interprofessional education program focused on a human sexuality course. Cleckley et al. [12]detailed the use of a design thinking process to identify discrepancies between expected interprofessional education competencies, curriculum implementation, and interprofessional collaboration. Gomutbutra et al. [13] discovered that both medical students and technology students perceived design thinking as beneficial to their careers and expressed satisfaction with its application in enhancing interprofessional education between technology and medicine for innovative elder pain care.

Point 2: State-of the art comparison is not present in study.

Response 2 : Thank you. We modify the whole article. And invite professional English editing experts to assist in editing the manuscript

Point 3: Paper is just like theory, no experimentation is visible at first sight.

Response 3 : We reinforce the paragraphs and explanations of our research methods

From line 163-189: We used action research as research method. Action research is a research method that combines investigation and problem-solving simultaneously [19]. In the field of education, it refers to systematic research that aims to bring about change and improvement in educational practice [20]. In the context of medical education, this research adopts the action research method to address a specific problem and enhance teaching. The focus of action research is not on developing theory or generalizable research findings but on designing research that addresses the characteristics of a specific problem in a particular time, place, and situation, while incorporating relevant educational theories [20]. Action research follows a systematic and cyclical inquiry process of planning, action, observation, and reflection. The following are the scenarios and steps for using action research in this study.

The educational course was led by three instructors and involved the participation of twenty individuals. We invites six doctors, seven nursing staffs, four case managers and three family members to conduct design thinking workshops. In addition to comprehensive lectures, the training incorporated a project that entailed the design of diabetes discharge preparation services. For this project, participants were divided into five teams. We conduct workshops using the five major processes of Design Thinking from Stanford University. Towards the conclusion of the course, feedback on the projects was solicited from both patients and medical specialists.

The diabetes network was chosen for this study due to the significance of diabetes care and management in healthcare. Comprised of healthcare professionals and family members, the network aimed to address challenges in diabetes care through the use of design thinking. Design thinking offered a systematic and collaborative approach to promote innovation, problem-solving, and patient-centered solutions. By applying design thinking principles, the network aimed to foster interdisciplinary collaboration, identify unmet patient needs, develop creative solutions, and enhance the quality of diabetes care and education.

Point 4: Presentation is not good.

Response 4 : Thank you. We modify the whole article. And invite professional English editing experts to assist in editing the manuscript

Reviewer 4 Report

Dear authors,

Your paper is an interesting one for readers, since design thinking is becoming more utilized in healthcare design. There are some revisions needed, however, to improve your paper's utility for the readers. Please see my comments on each section of your paper:

Introduction

The first sentence with COVID-19 does not fit anymore. It sounds like your focus is design thinking by interprofessional teams, beginning with diabetic patients. I would reframe your Introduction to describe the importance of design thinking for healthcare team discharge planning, and then explain why you focused your study on diabetic patients.

On page 2, I like your succinct description of design thinking. I wonder if it would be possible to include a table comparing traditional problem-solving to design thinking? I also need clarification for this statement on line 51: “It is crucial to go beyond the framework..” What framework are you referring to?

On page 3, you name the Stanford School of Design curriculum. Is this one design curriculum you’re referencing, or is this the approach you used? If it’s an example, I would state it as an example of how design thinking can be conducted.

Materials and Methods

Please provide a rationale for why you chose to use with the diabetes network? Who comprises the network? Why was design thinking needed by them?

Content in Results sounds like it belongs under Materials and Methods. I recommend creating a Materials and Methods table where you indicate key activities and tools used for each of the design stages. In this section, it would be helpful to readers to add brief descriptions of the tools you’ve referenced, such as the Persona Tool. If you have a workbook, agenda or other materials to aid replication of your approach, I recommend including more in-depth, detailed materials as supplementary text.

The Results should include feedback from participants and outcomes from the workshop.

The Discussion should tie your findings to other examples of design thinking in the literature. Were your outcomes and deliverables similar to other design thinking studies in the published literature?

Overall, there were few issues with readability. I would try to take out any narrative that is not needed. Tables are a good addition. 

Author Response

Response to Reviewer 4 Comments

Point 1: The first sentence with COVID-19 does not fit anymore. It sounds like your focus is design thinking by interprofessional teams, beginning with diabetic patients. I would reframe your Introduction to describe the importance of design thinking for healthcare team discharge planning, and then explain why you focused your study on diabetic patients.

Response 1: We have re-written the introduction. From line 33-68.

Point 2: On page 2, I like your succinct description of design thinking. I wonder if it would be possible to include a table comparing traditional problem-solving to design thinking? I also need clarification for this statement on line 51: “It is crucial to go beyond the framework..” What framework are you referring to?

Response 2: We have incorporated design thinking into our interprofessional education approach due to its distinct emphasis on the contributions of various disciplinary professions, rather than solely focusing on the problem at hand. By leveraging design thinking principles, we aim to foster collaboration and encourage the integration of diverse perspectives from different professions. This approach enables us to explore innovative solutions that encompass the expertise and insights of all team members, leading to more comprehensive and effective outcomes. Through the utilization of design thinking, we prioritize the collective wisdom and expertise of the interprofessional team to address complex healthcare challenges and provide holistic care to patients.

We also re-written description of design thinking from line 68-86: Traditionally, hospitals have utilized "case discussions" as a means to facilitate IPE. [5]. This approach focuses on individual expertise rather than under-standing the roles of others and fostering cooperative learning and interaction [6]. This study suggests that the educational approach of design thinking is more effective in enhancing the effectiveness of collaborative nursing than case discussions, allowing professionals, patients, and their families from various disciplines to brainstorm and promote patient-centered care methods. In the following paragraphs of this study, we will first introduce the connotation of design thinking, explaining that it is a process that utilizes the unique perspectives, experiences and skills of each participant, through cross-disciplinary cooperation and continuous improvement. This study uses the action research method and actually invites doctors, nursing staff, case managers and family members to conduct design thinking workshops. The research result is to find out an innovative and patient-centered discharge care model. [7].

Point 3: On page 3, you name the Stanford School of Design curriculum. Is this one design curriculum you’re referencing, or is this the approach you used? If it’s an example, I would state it as an example of how design thinking can be conducted.

Response 3: We follow you suggestion and explain at line 179-181.

Point 4: Please provide a rationale for why you chose to use with the diabetes network? Who comprises the network? Why was design thinking needed by them?

Response 4: We explain at line 182-188: The diabetes network was chosen for this study due to the significance of diabetes care and management in healthcare. Comprised of healthcare professionals and family members, the network aimed to address challenges in diabetes care through the use of design thinking. Design thinking offered a systematic and collaborative approach to promote innovation, problem-solving, and patient-centered solutions. By applying design thinking principles, the network aimed to foster interdisciplinary collaboration, identify unmet patient needs, develop creative solutions, and enhance the quality of diabetes care and education.

Point 5: Content in Results sounds like it belongs under Materials and Methods. I recommend creating a Materials and Methods table where you indicate key activities and tools used for each of the design stages. In this section, it would be helpful to readers to add brief descriptions of the tools you’ve referenced, such as the Persona Tool. If you have a workbook, agenda or other materials to aid replication of your approach, I recommend including more in-depth, detailed materials as supplementary text.

Response 5: We follow your suggestion and modify the content of result from line 257-267 and also make a new table (see table 1).

Point 6: The Results should include feedback from participants and outcomes from the workshop.

Response 6: We add the result at line 257-266: These solutions consisted of personalized dietary recommendations specifically designed for diabetes management on food ordering and delivery platforms. Additionally, a visually captivating health education picture book incorporating dynamic character dialogues and audio animations was developed. Furthermore, a proposal was presented for the production of a health education short film, complemented by both physical and electronic books. These mediums were strategically designed to deliver informative messages to patients and their families in captivating ways, offering comprehensive guidance throughout the patient care process before, during, and after discharge. The results indicate the potential of these solutions to enhance patient engagement and improve the effectiveness of diabetes care.

Point 7: The Discussion should tie your findings to other examples of design thinking in the literature. Were your outcomes and deliverables similar to other design thinking studies in the published literature?

Response 7: We add one more paragraph to discuss the findings from line 291-298: Our research aligns with previous studies in the literature, which have also shown that employing design thinking methodologies is effective in fostering interdisciplinary collaboration, promoting creative problem-solving, and improving patient care outcomes. For example, in healthcare and education, design thinking has been found to enhance interprofessional education programs and improve educational practices [11, 12 ,13]. These studies emphasize the importance of engaging diverse stakeholders, encouraging cross-disciplinary dialogue, and integrating design thinking principles to address specific challenges and develop innovative solutions.

Round 2

Reviewer 2 Report

The manuscript was improved significantly.

Author Response

Thank you for your affirmation. With the suggestions of all the reviewers, it is possible to make the article better.

Reviewer 3 Report

1. Improve abstract of the manuscript.

2. Use simple and concise statements.

3. Improve conclusion.

4. Research work is not compared with existing work clearly.

5. Grammar and typo error should be avoid.

Minor corrections required in grammar.

Author Response

We have revised the article again and made effort to correct any errors. 

Reviewer 4 Report

Thank you for the revisions you have made to date. There are some additional revisions needed before publication. Here are my recommendations:

1. Use consistent terminology throughout, particularly "multidisciplinary." For example: 

Line 60 on page 2. You mention “collaborative nursing.” I think you mean collaborative, multidisciplinary teamwork. Line 74, page 2. You use “cross-functional teams.” In the manuscript, you use “medical services” and “medical team.” Please use “multidisciplinary” when referring to the service, team or professionals. Readers may think you’re only discussing the role of medical professionals.

2. Please substitute "customers" with "service users." Customers is a business term. For example, line 72 on page 2. 

3. The design thinking process you mention has 5 stages. Please reference this process after a brief description of the 5 stages. It not clear why there are different references for individual stages if they all refer to one process you used. The stages are clear to follow except for stage 2. What is being defined? A problem that needs a solution? Lines 124-126 are not clear to me. Be specific about the key goal of each stage. You mention the use of case scenarios in stage 2, although earlier, you mention how design thinking is an alternative to case scenarios. It sounds like cases are still used, but integrated within a more extensive design process. Is this correct?

4. Line 110, page 3. What do you mean by “pain points”? Some readers might interpret this literally.

5. At the end of your overview of the design process, state the aim of your project and a rationale for your project. For example, was this a quality improvement project based on service user feedback for better diabetic services? It’s not clear why you did this project. Some of this information is under Methods, and you want your aim and goals/rationale to come before Methods. Be sure and use past tense for a project completed in 2020. Did you carry out all the steps in a one-day workshop? Is this typical? On line 174 of page 5, you state “ideation project.” This is confusing, since “ideate” is a stage of the design process. Perhaps be consistent by calling your work a design thinking project.

5. Methods

Did you really do action research? It sounds like you did a 1-day design thinking process that engages users and stakeholders. You might do better focusing on design thinking and describing the methods and tools you used for design thinking. I think your description of action research will confuse readers, particularly how you describe action research in your paper. For example, you don’t mention education until the Methods section  You refer to action research methods for education, and you describe what you did as an educational course on page 5, line 186. Although there may have been an educational component to the process, design thinking goes beyond education. Readers will be confused by your introduction of action research for education and your description of an educational course. I The Methods section, therefore, needs to be revised to reflect how you engaged users and stakeholders with the stages and tools of the design thinking process. 

6. Page 5, lines 194-196 belong in the Introduction or Background.

7. The table of tools is a great addition. Please move information about the tools used in each stage up to the Methods section. These were tools you used for your design thinking process.

8. Results needs to report outcomes from your 1-day workshop.

9. Discussion needs to compare your learnings from the design thinking process to other design thinking projects. Was there ‘value-add’ from using the design thinking approach versus the more traditional case-based approach you describe earlier on? You have some of the Discussion content under Conclusions.

10. Conclusions should simply summarize your key learnings from the project. 

Overall, the use of design thinking for diabetes management will be an interesting one for readers, but the revisions above will strengthen your paper. 

There are some issues with tenses (using present versus past). There are hyphens throughout that don't belong there. 

Author Response

Response to Reviewer 4 Comments

Point 1: Line 60 on page 2. You mention “collaborative nursing.” I think you mean collaborative, multidisciplinary teamwork. Line 74, page 2. You use “cross-functional teams.” In the manuscript, you use “medical services” and “medical team.” Please use “multidisciplinary” when referring to the service, team or professionals. Readers may think you’re only discussing the role of medical professionals.

Response 1: Thank you. We changed "collaborative nursing" to ""collaborative teamwork"(line 59). We also changed " cross-functional teams " to "" multidisciplinary ".(line73)

-----------------------------------------------------------------------------------------------------------------

Point 2: Please substitute "customers" with "service users." Customers is a business term. For example, line 72 on page 2.

Response 2: We have revised the 'customers' in the entire article to' service users'.(line 71-74)

-----------------------------------------------------------------------------------------

Point 3: The design thinking process you mention has 5 stages. Please reference this process after a brief description of the 5 stages. It not clear why there are different references for individual stages if they all refer to one process you used. The stages are clear to follow except for stage 2. What is being defined? A problem that needs a solution? Lines 124-126 are not clear to me. Be specific about the key goal of each stage. You mention the use of case scenarios in stage 2, although earlier, you mention how design thinking is an alternative to case scenarios. It sounds like cases are still used, but integrated within a more extensive design process. Is this correct?

Response 3: Stage 1 is to identify the persistent problems faced by users, while Stage 2 is to select the most prioritized problem to be solved from these problems, which we call the “Define”.We modify the line 124-126. “After defining the problem, users or stakeholders can be consulted to validate the accuracy of the identified problem statement. In Define Stage, there is no need to ask experts to directly provide an action plan, but rather to enter Ideate Stage and find more action plans through collective creative thinking.”

I value your input and I am grateful for the opportunity to address any issues you have raised, and look forward to resolving this matter to your satisfaction.

-----------------------------------------------------------------------------------------

Point 4: Line 110, page 3. What do you mean by “pain points”? Some readers might interpret this literally.

Response 4: We modify the sentence at line 110. “…to facilitate innovators in gaining a comprehensive understanding of users' experiences and persistent problems with the service that can inconvenience them,…”. We have revised the 'pay points' throughout the article to' persistent problems'.

-----------------------------------------------------------------------------------------

Point 5: 5 At the end of your overview of the design process, state the aim of your project and a rationale for your project. For example, was this a quality improvement project based on service user feedback for better diabetic services? It’s not clear why you did this project. Some of this information is under Methods, and you want your aim and goals/rationale to come before Methods. Be sure and use past tense for a project completed in 2020. Did you carry out all the steps in a one-day workshop? Is this typical? On line 174 of page 5, you state “ideation project.” This is confusing, since “ideate” is a stage of the design process. Perhaps be consistent by calling your work a design thinking project.

Response 5: We follow your suggestion and modify the line 174.:“We fully implemented the five steps of design thinking. The final design thinking result is to invite the Deputy Director of the Nursing Department and family members to evaluate whether the plan is feasible and valuable. We hope that the proposals proposed by various groups can be used to improve the health care of diabetes patients in the future.” This article aims to highlight the immense potential of continued education in revolutionizing curriculum design by fostering creativity, embracing innovative business models, and adopting highly efficient approaches. By delving into these transformative concepts, we strive to unleash the true power of education, empowering educators and learners alike to envision and implement groundbreaking ideas that shape the future of education. With a focus on creativity and progressive methodologies, we hope this article aspires to inspire educational professionals to push boundaries, embrace change, and pave the way for an educational landscape that fosters ingenuity and unlocks untapped potential.

-----------------------------------------------------------------------------------------

Point 6: Did you really do action research? It sounds like you did a 1-day design thinking process that engages users and stakeholders. You might do better focusing on design thinking and describing the methods and tools you used for design thinking. I think your description of action research will confuse readers, particularly how you describe action research in your paper. For example, you don’t mention education until the Methods section  You refer to action research methods for education, and you describe what you did as an educational course on page 5, line 186. Although there may have been an educational component to the process, design thinking goes beyond education. Readers will be confused by your introduction of action research for education and your description of an educational course. I The Methods section, therefore, needs to be revised to reflect how you engaged users and stakeholders with the stages and tools of the design thinking process.

Response 6: Although we have conducted extensive action research, the limited space in this Brief Report prevents us from providing a comprehensive presentation of the research process and analysis. However, we intend to publish these studies in the near future, allowing for a detailed exploration of our findings.

But we still follow your suggestion and modify the Method section.

(line 177-188)

“The continuing education program was led by three instructors and involved the participation of twenty individuals. We invite six doctors, seven nursing staffs, four case managers and three family members to join this workshop. All participants were divided into five teams. Throughout the activity, participants engaged in numerous discussions, documenting their ideas on sticky notes or posters. We have preserved all of these files as part of our study. Furthermore, we diligently recorded the shared content from each team during the event. To gather additional insights, we developed a reflection questionnaire to capture participants' thoughts and suggestions regarding their participation. These materials serve as valuable research text, enabling us to analyze and draw conclusions from the study.”

-----------------------------------------------------------------------------------------

Point 7: The table of tools is a great addition. Please move information about the tools used in each stage up to the Methods section. These were tools you used for your design thinking process.

Response 7: We move table 1 to Methods section.

-----------------------------------------------------------------------------------------

Point 8: Results needs to report outcomes from your 1-day workshop.

Response 8: We report at line 161, “This one-day, eight-hour program, held at a teaching hospital in eastern Taiwan.”  

-----------------------------------------------------------------------------------------

Point 9:Discussion needs to compare your learnings from the design thinking process to other design thinking projects. Was there ‘value-add’ from using the design thinking approach versus the more traditional case-based approach you describe earlier on? You have some of the Discussion content under Conclusions.:

Response 9: We modify the conclusion from line 280-287. “Design thinking is a methodology for generating innovative solutions to complex problems, whereas case studies are research methods used to examine specific instances or phenomena in detail. Both approaches offer valuable insights and have their unique strengths in different research and problem-solving contexts.

Our research can be used to illustrate that the process of problem-solving should not be pursued in isolation; instead, it necessitates the cultivation of interdisciplinary literacy and efficient teamwork.”

Point 10: Conclusions should simply summarize your key learnings from the project.

Response 10: We add simply summarize of key learnings from project(line 301-308). “We finally summarize two possible methods: personalized dietary recommendations for diabetes management on food ordering and delivery food platforms. While these methods require further implementation and validation for their effectiveness, the process itself has been invaluable. It has allowed participants to identify user needs, explore problems from diverse perspectives, and collaboratively reach consensus on problem-solving. As a result, we strongly believe that design thinking is highly suitable for medical education as it fosters empathy and enhances problem-solving abilities. By employing design thinking principles, we can cultivate a more patient-centered approach to healthcare.”

-----------------------------------------------------------------------------------------